

# A restructured and updated global soil respiration database (SRDB-V5)

Jinshi Jian[1], Rodrigo Vargas[2], Kristina Anderson-Teixeira[3,4], Emma Stell[2], Valentine Herrmann[3], Mercedes Horn[5], Nazar Kholod[1], Jason Manzon[6], Rebecca Marchesi[3], Darlin Paredes[7], and Ben Bond-Lamberty[1*]

*Correspondence to: Ben Bond-Lamberty (bondlamberty@pnnl.gov)*

[1] Pacific Northwest National Laboratory, Joint Global Change Research Institute at the University of Maryland–College Park, 5825 University Research Court, Suite 3500, College Park, MD 20740, USA

[2] Department of Plant and Soil Sciences, University of Delaware, Newark, DE, USA

[3] Conservation Ecology Center; Smithsonian Conservation Biology Institute; Front Royal, VA, USA

[4] Center for Tropical Forest Science-Forest Global Earth Observatory; Smithsonian Tropical Research Institute; Panama, Republic of Panama

[5] University of Vermont, Rubenstein School of Environment and Natural Resources

[6] University of Maryland

[7] Georgetown University, School of Foreign Service

## Abstract

Field-measured soil respiration ($R_S$, the soil-to-atmosphere $CO_2$ flux) observations were compiled into a global soil respiration database (SRDB) a decade ago, a resource that has been widely used by the biogeochemistry community to advance our understanding of $R_S$ dynamics. Novel carbon cycle sciences questions require updated and augmented global information with better interoperability among datasets. Here, we restructured and updated the global $R_S$ database to version SRDB-V5. The updated version has all previous fields revised for consistency and simplicity, and it has several new fields to include ancillary information (e.g., $R_S$ measurement time, collar insertion depth, collar area). The new SRDB-V5 includes published papers through 2017 (800 independent studies) where total observations increased from 6633 in SRDB-V4 to 10366 in SRDB-V5. The SRDB-V5 features more $R_S$ data published in Russian and Chinese scientific literature, has an improved global spatio-temporal coverage, and improved global climate-space representation. We also restructured the database so that it has stronger interoperability with other datasets related to carbon-cycle science. For instance, linking SRDB-V5 with an hourly timescale global soil respiration database (HGRsD) and an open community database for continuous soil respiration and other chamber flux data (COSORE) enables researchers to explore new questions. The updated SRDB-V5 aims to be a data framework for the scientific community to share seasonal to annual field $R_S$ measurements, and it provides opportunities for the biogeochemistry community to better understand the spatial and temporal variability of $R_S$, its components, and the overall carbon cycle.

The database can be downloaded at https://github.com/bpbond/srdb and ORNL DAAC [Submitted].

All data and code to reproduce the results in this study can be found at: Jian, Jinshi, Bond-Lamberty, Ben. (2020). jinshijian/ESSD: SRDB-V5 first release (Version v1.0.0) [Data set]. Zenodo. http://doi.org/10.5281/zenodo.3876443.

**Keywords**: SRDB, soil respiration, database, interoperability, reusability

## 1. Introduction

**Soil respiration ($R_S$), the soil surface to atmosphere $CO_2$ flux, is one of the largest carbon fluxes between the**
**terrestrial land surface and atmosphere** (Luo and Zhou, 2010). **The majority of $R_S$ is released by soil
microbial/fauna (heterotrophic respiration) and plant root respiration (autotrophic respiration).** Soils hold a
large amount (>2000 Pg C to 1 m depth) of carbon, more than the total of carbon stock in the atmosphere and
aboveground plants (Batjes, 2016; Tarnocai et al., 2009). Thus, its C efflux to the atmosphere has significant
implications for our understanding of ecosystem- to global-scale biogeochemical cycling. For better monitoring soil
carbon dynamics as well as to investigate how soil carbon responds to global climate change, it is important to
measure $R_S$ across different vegetation types and climate conditions.

**Many field experiments have been conducted in recent decades to measure $R_S$ in different climate conditions
and vegetation types** (Bond-Lamberty and Thomson, 2010b; Davidson et al., 1998; Raich and Potter, 1995)**.**
However, the resulting estimates of seasonal to annual $R_S$ fluxes are scattered throughout the scientific literature in a
variety of formats. Therefore, compiling past $R_S$ measurements together into a standardized data framework to
support synthesis analysis is very important to advance carbon cycle science.

**Published site scale $R_S$ measurements across the globe have been compiled and standardized into global soil**
**respiration databases to support synthesis studies, macro-to-global scale $R_S$ estimates, and soil carbon
response to climate change investigation** (Bond-Lamberty and Thomson, 2010a; Raich and Schlesinger, 1992)**.**
Schlesinger (1977) compiled one of the earliest listings of $R_S$ estimates from diverse ecosystems. Raich and
Schlesinger (1992) subsequently integrated $R_S$ from published papers which covered 13 ecosystems, and developed
a simple linear model between $R_S$ and climate factors (i.e., temperature and precipitation), estimating global $R_S$ to be
$68 \pm 4$ Pg C yr$^{-1}$. Later, more $R_S$ measurements (especially measured using Infra-Red Gas Analyzers (IRGA method)
were added and the global $R_S$ was updated to 76-81 Pg C yr$^{-1}$ (Raich et al., 2002; Raich and Potter, 1995). In 2010,
Bond-Lamberty and Thomson (2010a) compiled a comprehensive global soil respiration database (SRDB) and this
database was released for public usage. The SRDB contains annual and seasonal $R_S$ measurements, ancillary carbon
pools and fluxes (e.g., gross primary production, net primary production, ecosystem respiration), response of $R_S$ to
temperature and moisture (i.e., model parameters to describe the relationship between $R_S$ and temperature and
moisture), and sites' background information (e.g., latitude, longitude, elevation, mean annual temperature, mean
annual precipitation) (Bond-Lamberty and Thomson, 2018, 2010a). With more IRGA-based $R_S$ measurements added
and alkaline-based measurements excluded, Bond-Lamberty and Thomson (2010b) estimated the global $R_S$ to be 98
$\pm 12$ Pg C yr$^{-1}$ and estimated that global $R_S$ was increasing at a rate of 0.1 Pg C yr$^{-2}$. The SRDB has been widely
used in the past decade since the first version was published (Bond-Lamberty and Thomson, 2010a), and to date it
has been cited 359 times (searched in Google Scholar on 5/20/2020) but its use continues to increase (Figure 1).

**The SRDB of Bond-Lamberty and Thomson (2010) however only recorded seasonal to annual $R_S$ fluxes,
hindering analyses at finer temporal resolutions.** Based on the SRDB, Jian et al. (2018c) collected SRDB studies
reporting diurnal $R_S$ and compiled these into an global hourly soil respiration database (HGRsD). Similarly, Jian et
al. (2018a) further collected detailed monthly/daily time scale $R_S$ measurements into a global monthly/daily soil
respiration database (MGRsD). More recently, Bond-Lamberty et al. (submitted) have built a database (COSORE)
of continuous (typically half-hourly or hourly) datasets from globally-distributed sites. With these different-
timescale databases, $R_S$ temporal variability, and its time-related driving processes and uncertainties, can be
analyzed (Jian et al., 2018a, 2018b, 2018c). There is still a need to improve interoperability among $R_S$ databases to
expand available information, improve database usage, and to advance our understanding of $R_S$ dynamics across
multiple spatial and temporal scales.

**In approaching a decadal reworking of the SRDB, we envisioned that it required improvements to increase**
**its usage across different disciplines.** Some important information (e.g., collar area, collar insertion depth, $R_S$





measure time, soil temperature, soil moisture, soil temperature measure depth, and soil moisture measure depth) was not included in the older versions (hereafter named SRDB-V1 to SRDB-V4), and thus important questions such as whether $R_S$ survey time (Cueva et al., 2017), collar insertion depth (Heinemeyer et al., 2011), and/or how collar cover area affected $R_S$ measurements accuracy could not be addressed. In addition, SRDB-V4 included data mainly published in English (~98%), while data published in other languages (~2%) were rarely included (Epule, 2015). Some metadata such as manipulation/treatments and measurement method were not standardized and thus were difficult to use in subsequent meta-analyses. For instance, the attempt to link SRDB to the Forest Carbon Database (ForC) showed that the old SRDB structure required modification before it can be linked with ForC (Anderson-Teixeira et al., 2018a, 2018b). Finally, information about how heterotrophic ($R_H$) and autotrophic respiration ($R_A$) respond to environmental conditions (i.e., temperature and soil moisture) was not included.

**The older SRDB followed certain data integration principles, including inclusion criteria, database structure design, and quality control** (Bond-Lamberty and Thomson, 2010a)**, but improvements could be made.** We have updated it to a new version (hereafter named SRDB-V5) following FAIR protocols (i.e., Findable, Accessible, Interoperable, and Reusable) (Wilkinson et al., 2016). This has been accomplished by 1) restructuring SRDB and improving its interoperability so that data from SRDB-V5 can more easily be linked to external datasets; 2) separating the $R_S$, $R_H$, and $R_A$ responses to temperature and soil moisture functions into a separate file to simplify the database and improve its reusability; 3) adding collar area, collar insertion depth, and $R_S$ measurement time information to SRDB-V5; 4) collecting more $R_S$ data published in Russian and Chinese scientific literature; 5) updating $R_S$ records available throughout the world from recently published literature (until 2017); and 6) improving the metadata description. We hope that these efforts will significantly improve the future interoperability and reusability of SRDB-V5.

## 2. Methods

### 2.1 Soil respiration database restructuring

**We restructured the SRDB for easier data collection and quality control.** The previous global $R_S$ database versions (SRDB-V1 to SRDB-V4) mainly included 2 files: a "studies" file, which recorded the detailed metadata for all published papers examined by the SRDB; and a "data" file, which stores all the $R_S$ data, a variety of ancillary site, soil, and carbon cycle data (e.g., GPP, NPP, ecosystem respiration), and related background information such as site location, ecosystem type, and management (Bond-Lamberty and Thomson, 2010a). In SRDB-V5 the "studies" file remains unchanged, but the "data" file is now separated into two files: "srdb-data" and "srdb-equations". This simplifies the structure of the former, while moving all the "Response of $R_S$ to temperature and moisture" columns in the SRDB to the latter.

### 2.2 Metadata
**We standardized the background information of SRDB-V5.** Most of the metadata are described by Bond-Lamberty and Thomson (2010a), and here we only describe new added columns or metadata with updates (Table 1 to Table 3). We added five columns (i.e., *Site_ID*, *Collar_height*, *Collar_depth*, *Chamber_area*, *Time_of_day*) in SRDB-V5. Four columns (*Rs_max*, *Rs_maxday*, *Rs_min*, *Rs_minday*) were deleted (Table 1) because they were rarely reported and had not been used by the community in the past ten years. In the *Quality_flag* column, we added two more flags related to $R_S$-temperature equations: Q15 means the equation was developed based on seasonal $R_S$ data rather than covering at least a whole year, and Q16 notes that there is a soil water content (SWC) component within the reported equation (Table 1).



**For many analyses SRDB needs to be connected with other datasets, and a unique observation ID is essential for this process.** In the SRDB-V5, we added a "*Site_ID*" column to guarantee a unique ID for each *Rs_annual* observation within a study, enabling users to easily link SRDB-V5 records with external data such as MGRsD and HGRsD. The *Site_ID* is in the form of 'CC-RC-IC', where CC is the ISO Alpha-2 country code (https://www.nationsonline.org/oneworld/country_code_list.htm), RC is region code (state/province), and IC is

identity code. Country code and region code are always present, but some studies report only one annual $R_S$ value, and thus IC may or may not be present.

**We standardized the coding of experimental manipulation, collapsing the previous *ad hoc* categories into a smaller set of standardized terms. This decreased the number of unique *Manipulation* field values from 689 to**

**276.** We used the following criteria to simplify the manipulation in SRDB-V5: 1) Measurements from no-treatment (i.e., control) were categorized as "None", 2) manipulation names were standardized (e.g., "clipping", "clip", and "clipped" are now all standardized as "Clip"), 3) we used the manipulation level to further describe the difference within a specific manipulation (e.g., "Litter manipulation" could have "double litter", "50% litter removal", "100% litter removal"). With manipulation standardized, scientists can further analyze how manipulation affects $R_S$. For

instance, comparing $R_S$ measurements from the "$CO_2$" group (i.e., elevated $CO_2$ concentration treatment) with "None" (i.e., control) enables researchers to analyze how $R_S$ responds to $CO_2$ concentration increase caused by $CO_2$ released from fossil fuel combustion. Similarly, data from the "Warm" and "Precipitation amount change" groupings will enable scientists to more easily explore how soil carbon responds to global climate change. Barba et al. (2018) suggested that bias could arise from measurements made in "hot-spots", and groupings such as "Ant

mound" and "High N" facilitate data interpretation and analyses regarding "hot-spots".

**We also standardized the $R_S$ measurement method (the *Meas_method*) and $R_S$ partition method (*Partition_method*) fields.** Measurement method was grouped into 9 types (Table 2) and the partition method was grouped into 8 types (Table 3). With these changes, scientists can more easily investigate whether different measure

methods affect $R_S$ results, as well as whether different partition methods affect $R_H$ and $R_A$ partitioning.

### 2.3 Soil respiration database update

**We updated the SRDB-V5 so that it has temporal coverage to 2017, and made an effort to collect $R_S$ data published in Russian and Chinese literature to be more inclusive and expand its spatial coverage.** Papers published in English are the majority (~98%) of sources in SRDB, while papers published in other languages are

rarely included (Bond-Lamberty and Thomson, 2018, 2010a). This reflects the dominance of English as the language of international science, but there are some data available from the Russian-language literature, representing data from a large area (Russia represents ~11% of the terrestrial land surface) and a variety of climate types and vegetation types. In addition, in MGRsD and HGRsD, there were some Chinese-language papers or recently-published papers (103 studies, ~5% of the total studies in SRDB-V5), which were not included by SRDB.

Now we have compiled data from those papers into SRDB-V5.

### 2.4 Data quality control

**We developed an R** (R Core Team, 2019) **script to perform data quality and consistency checks.** For example, the *Latitude* and *Longitude* fields should be within -90 to 90 and -180 to 180 degrees, respectively; whenever they are out of these ranges, a warning is raised. For details about the data constraints used to check each column in

SRDB-V5 please see the 'srdb_check.R' script, which is available in the GitHub repository and as part of every release download (https://github.com/bpbond/srdb/releases). This script is also run on all *pull requests* to the Github repository, which enables us to flag data-quality problems before changes are made to the database.





**2.5 Data coverage analysis**


**We compared mean annual temperature (MAT) and mean annual precipitation (MAP) of sites from SRDB with the global MAT and MAP to test the representation of the SRDB.** We connected the sites from SRDB with external climate data (Willmott and Matsuura, 2001) through latitude and longitude, and obtained MAT and MAP. Barren area was masked according to the MODIS landcover (Friedl et al., 2002). Climate region was retrieved from the climate Köppen classification (Peel et al., 2007). We also obtained IGBP vegetation classification of the SRDB sites by connecting IGBP classification data (IGBP, 1990); vegetation was grouped into Agriculture, Arctic, Desert,

tropical forest (Tropic FOR), temperate & boreal forest (T&B FOR), Grassland, Savanna, Shrubland, Urban, and Wetland. If the MAT and MAP distribution of SRDB sites is similar to that of global MAT and MAP distribution, it should mean that the SRDB better represents the global flux $R_S$ distribution as well. We also assume that as data sample size increases, the new database (e.g., SRDB-V5) should improve its representation compared with the older version (e.g., SRDB-V1). We tested the representation of sites in different vegetation types (IGBP, 1990).


**Table 1.** Summary of metadata updates in SRDB-V5 compared with the old version SRDB-V4.

| Column | Description | Comments |
|---|---|---|
| *Site_ID* | CC-RC-IC (country code - region code - identity code) | Added in SRDB-V5 |
| *Collar_height* | Total height of collar | Added in SRDB-V5 |
| *Collar_depth* | Depth of collar inserted into soil (always < Collar_height) | Added in SRDB-V5 |
| *Chamber_area* | Area of collar covering the surface | Added in SRDB-V5 |
| *Time_of_day* | $R_S$ survey time (e.g., 8to12 represents $R_S$ measured from 8:00 to 12:00, local time; 0to24 stands for continuous measurement) | Added in SRDB-V5 |
| *Rs_max* | Maximum $R_S$ rate in a year | Deleted in SRDB-V5 |
| *Rs_maxday* | Day of year Rs_max recorded | Deleted in SRDB-V5 |
| *Rs_min* | Minimum $R_S$ rate in a year | Deleted in SRDB-V5 |
| *Rs_minday* | Day of year Rs_min recorded | Deleted in SRDB-V5 |
| *Quality_flag* | Q15: equation simulated based on seasonal rather than annual data; Q16: Equation with SWC component | Updated in SRDB-V5 |
| *Manipulation* | Decreased from 689 unique values to 276 after being standardized | Standardized in SRDB-V5 |
| *Measure_method* | See Table 2 | Standardized in SRDB-V5 |
| *Partition_method* | See Table 3 | Standardized in SRDB-V5 |



**Table 2.** Summary of standardized measurement method (*Meas_method*) in SRDB-V5.

| Meas_method | Number of rows (n) | Comments |
|---|---|---|
| IGRA | 7734 | Type of Infrared gas analyzer (e.g., LICOR 8100A) |
| Gas chromatography | 1268 | Take gas samples in the field, and measure $CO_2$ concentration back in the laboratory to determine soil respiration rate |
| Alkali absorption | 910 | Using alkali absorption of $CO_2$ to determine soil respiration rate |
| Not reported | 238 | Measure method not reported in the study |
| EC | 88 | Eddy covariance |
| Gradient | 83 | Measure $CO_2$ concentration at different soil depth and calculate soil respiration rate based on gas diffusion law |
| Equation | 15 | Indirectly calculate soil respiration rate (e.g., through relationship between soil respiration and GPP) |
| Isotope | 3 | Determine soil respiration rate using isotope (e.g., $C^{13}$) |
| Unknown | 27 | None of above |

**Table 3.** Summary of standardized partition method (*Partition_method*) in SRDB-V5.

| Partition_method | Number of rows (n) | Comments |
|---|---|---|
| Comparison | 150 | Separating soil respiration into heterotrophic and autotrophic components by comparing with e.g., bare, clearcut, gap, or clip site |
| Exclusion | 1121 | Removing roots by trenching, deeply insert PVC pipe etc. |
| Extraction | 180 | Directly measure respiration from root to get autotrophic respiration |
| Girdling | 23 | Strips the stem bark to the depth of xylem, and measure respiration few months later to get the heterotrophic respiration |
| Isotope | 68 | Separating heterotrophic and autotrophic respiration through isotope labeling |
| Model | 49 | Separating heterotrophic and autotrophic respiration through a relationship (e.g., the relationship reported by (Bond-Lamberty et al., 2004)) |
| TBCA | 16 | Determining heterotrophic and autotrophic respiration through total belowground carbon allocation calculation |
| Other | 122 | None of above |

## 3. Results

**The sample size of SRDB-V5 is much larger compared with older versions.** Collecting $R_S$ measurements from newly published literature (until 2017) greatly improves the total number of observations in the database (increased from 6633 to 10366) in SRDB-V5, but only somewhat improved its spatial coverage (Figure 2). The northern hemisphere mid-latitude regions, where SRDB-V4 has the most $R_S$ sites, had the largest $R_S$ increase in SRDB-V5 as well (blue dots in Figure 2). Adding literature in Chinese did not substantially improve the spatial coverage either,

possibly because more and more $R_S$ measurements in China have been published in English scientific literature. However, most sites in China are from the eastern part of the country, and measurements from western China, if available, will be important to include in future SRDB updates. We collected ~50 papers published in Russian, but only 14 of them (~0.7% of total studies of all languages in SRDB-V5) met the criteria (see (Bond-Lamberty and Thomson, 2010a) for details) and were included in the database. This small number of papers nonetheless

substantially improved the database's spatial coverage of the Russian landmass (orange circles in Figure 2).

**MAT and MAP distribution of SRDB sites are very similar to global distribution in Agriculture, Forest, and Grassland regions, indicating good representativeness of SRDB sites in these three vegetations (Figures 3 and 4)**. For Shrublands, sites in the oldest versions of the database (e.g., SRDB-V4) did not represent the global

distribution well, but this distribution was greatly improved as more $R_S$ measurements were included in SRDB-V5 (Figure 3). Sites from other vegetation types, however, were less representative of the corresponding global climate space, with barren lands were masked out (Figure 3, right panel). More specifically, Arctic sites in SRDB have relatively narrow MAT and MAP coverage compared with the global Arctic MAT and MAP distribution, probably because many regions in the Arctic are covered by snow all year round, and thus it is difficult to measure $R_S$ in those

sites (Virkkala et al., 2019). Desert SRDB sites have lower MAT but higher MAP than the global distribution, probably because: 1) the disproportionate amount of samples in temperate regions (Figure 2) means that most sample in deserts are likely from wetter deserts; 2) the Sahara has low MAP and high MAT and covers a large area of the world, but few studies were conducted there so that area of the world may simply represents the bias; and 3) many "deserts" that have been studied are in relatively close proximity with urban developments (e.g., southwestern

USA, southern Europe) and those deserts are not as harsh nor extensive as the Sahara. Urban and Savanna sites in SRDB had lower MAT compared to their global distribution, probably because many tropical cities and savannas in South America, Asia, and Africa were rarely measured (Jian et al., 2020; Martin et al., 2012). We suggest that papers written in other languages, especially those in Portuguese, Spanish, and French could potentially increase the $R_S$ measurements in South America and Africa.

**Adding new measurements did not change the distribution of annual $R_S$ or seasonal $R_S$ (Figure 5), although** as noted above SRDB-V5 has significantly more total observations than SRDB-V4. Seasonal $R_S$ (growing, dry, wet, spring, summer, autumn, and winter season $R_S$) were similar in the SRDB-V5 compared to SRDB-V4 (Figure 5). We suspect that new $R_S$ measurements are collected disproportionately from the same regions as previously

sampled, and thus future studies should focus more on those regions with less data. For the future SRDB update, measurements from the Southern hemisphere, Desert, Arctic, and tropical forests, if available, will be important to include.



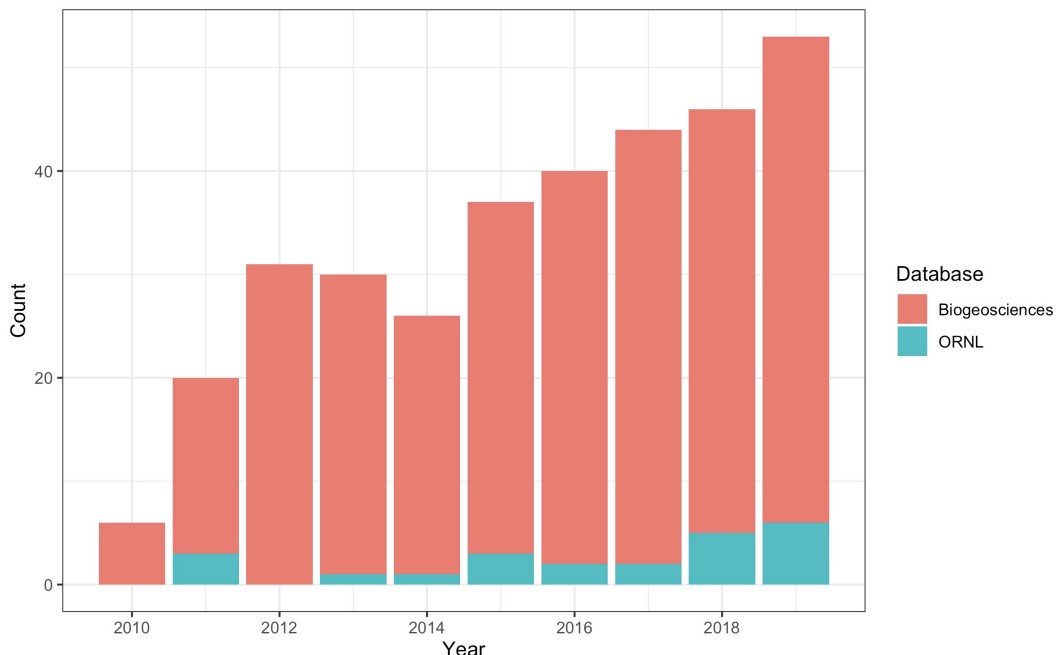


**Figure 1. Summary of studies citing the global soil respiration database (SRDB) between 2010 and 2019.** More and more studies are using SRDB since the first version (SRDB-V1) was published (Bond-Lamberty and Thomson, 2010a).



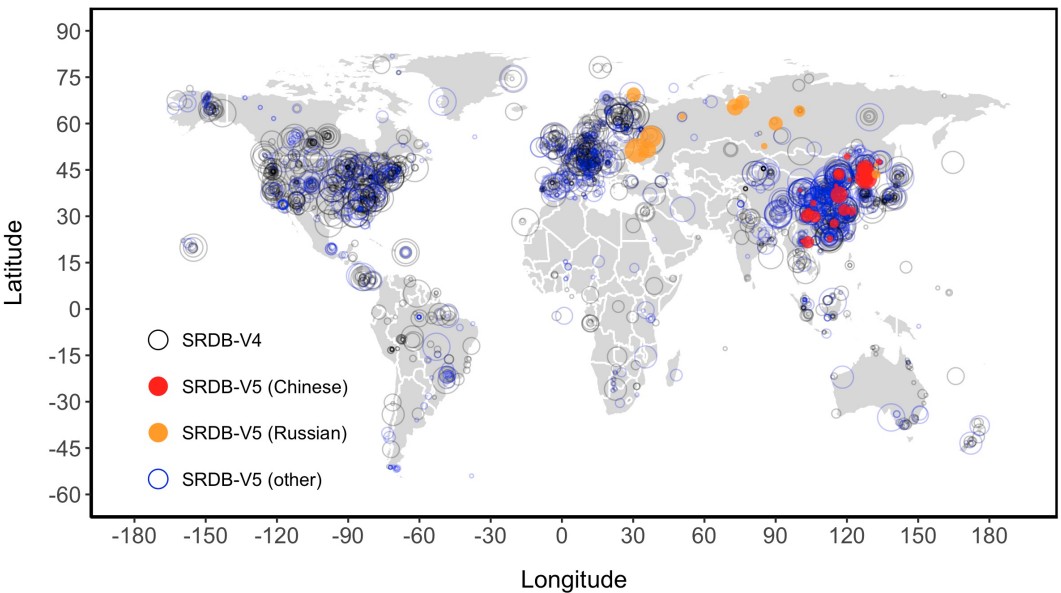


**Figure 2. Spatial distribution of soil respiration (R$_S$) sites.** The gray circles are R$_S$ sites from the fourth version of global soil respiration database (SRDB-V4, n=1584); the red dots are sites from the literature published in Chinese and added in the fifth version of global soil respiration database (SRDB-V5, n=41), the orange dots represent sites from the literature published in Russian and added in SRDB-V5 (n=16); the blue dots are sites from the literature published in other languages (mainly in English) and added in the SRDB-V5 (n=840). The size of circles represents the sample size at each measurement site (i.e., bigger circles represent more data).


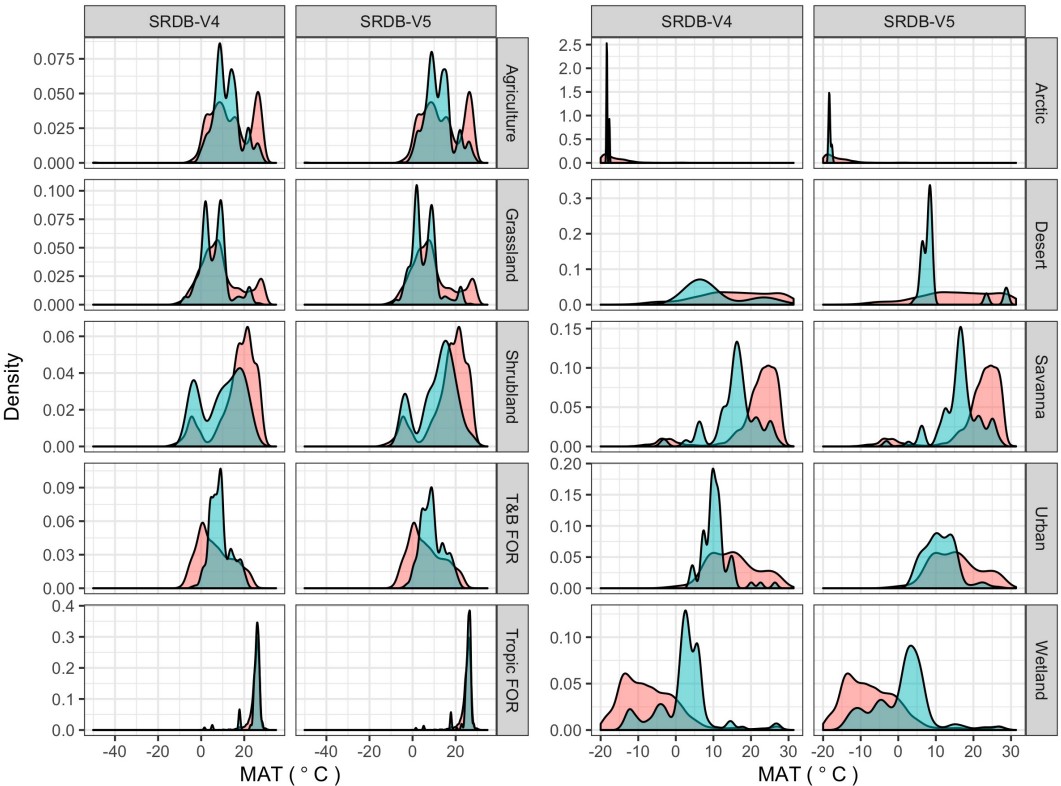

**Figure 3. Comparison of mean annual temperature (MAT, ℃) in the globe (in red) vs. MAT from the sites in the global soil respiration database (SRDB, in teal) by the vegetation types.** SRDB-4 represents the older SRDB released in 2018 and SRDB-V5 represents the newest SRDB published in 2020. Data from SRDB cover ten vegetation types (Agriculture, Arctic, Desert, tropical forest (Tropic FOR), temperate and boreal forest (T&B FOR), Grassland, Savanna, Shrubland, Urban, and Wetland). Comparing the forth version (SRDB-V4) to the newest version (SRDB-V5), MAT values of Agriculture, Forest, and Grassland sites generally well represent the global MAT; in contrast, MAT from Shrubland sites in the database did not well represent global means in the older SRDB-V4, but their representation significantly improved in the newest SRDB-V5; for other vegetation types (Arctic, Desert, Savanna, Urban, and Wetland (including peatland) in the right panel), the MAT of the database sites do not well represent the global MAT distribution. Note that the Barren region was masked using MODIS landcover data.

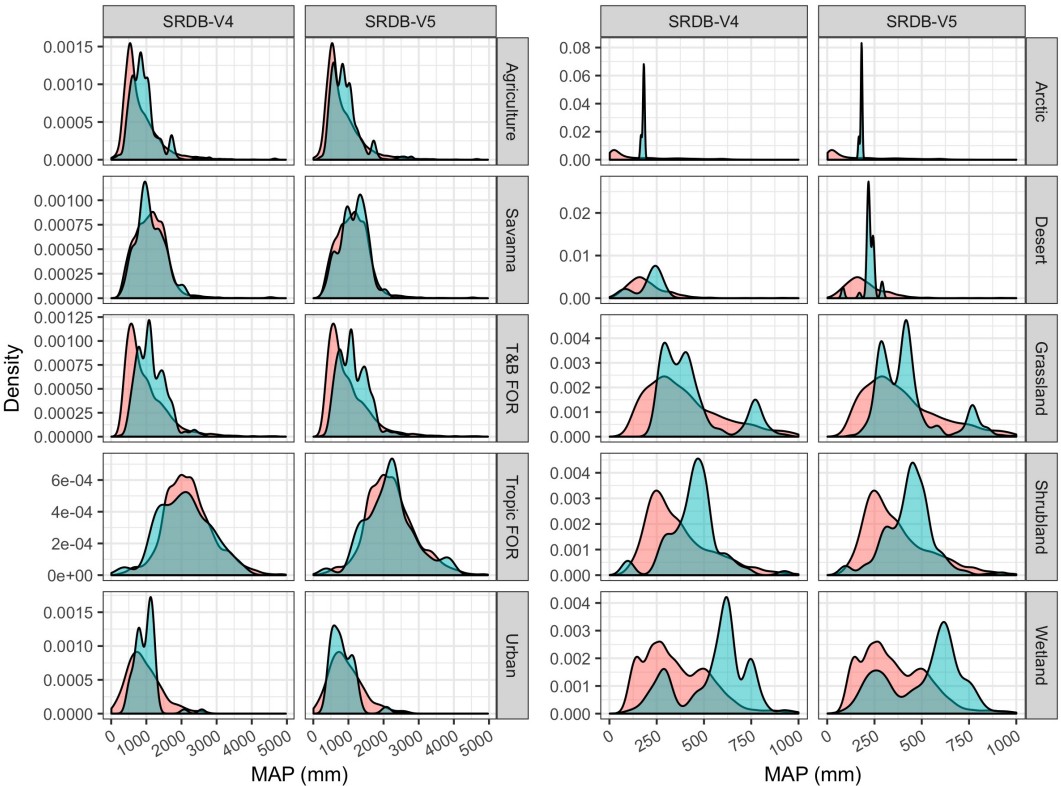

**Figure 4. Comparison of mean annual precipitation (MAP, mm) in the globe (in red) vs. MAP from the sites in the global soil respiration database (SRDB, in teal) by the vegetation types.** SRDB-V4 is the older SRDB published in 2018 and SRDB-V5 is the newest SRDB published in 2020. Data from SRDB covered ten vegetation types (see Figure 3). Sites from Agriculture, Savanna, Forest, and Urban generally well represent the global MAP (left panel), while sites from Arctic, Desert, Grassland, Shrubland, and Wetland (including peatland) do not have a good MAP representation (right panel). Note that the Barren region was masked using MODIS landcover data.




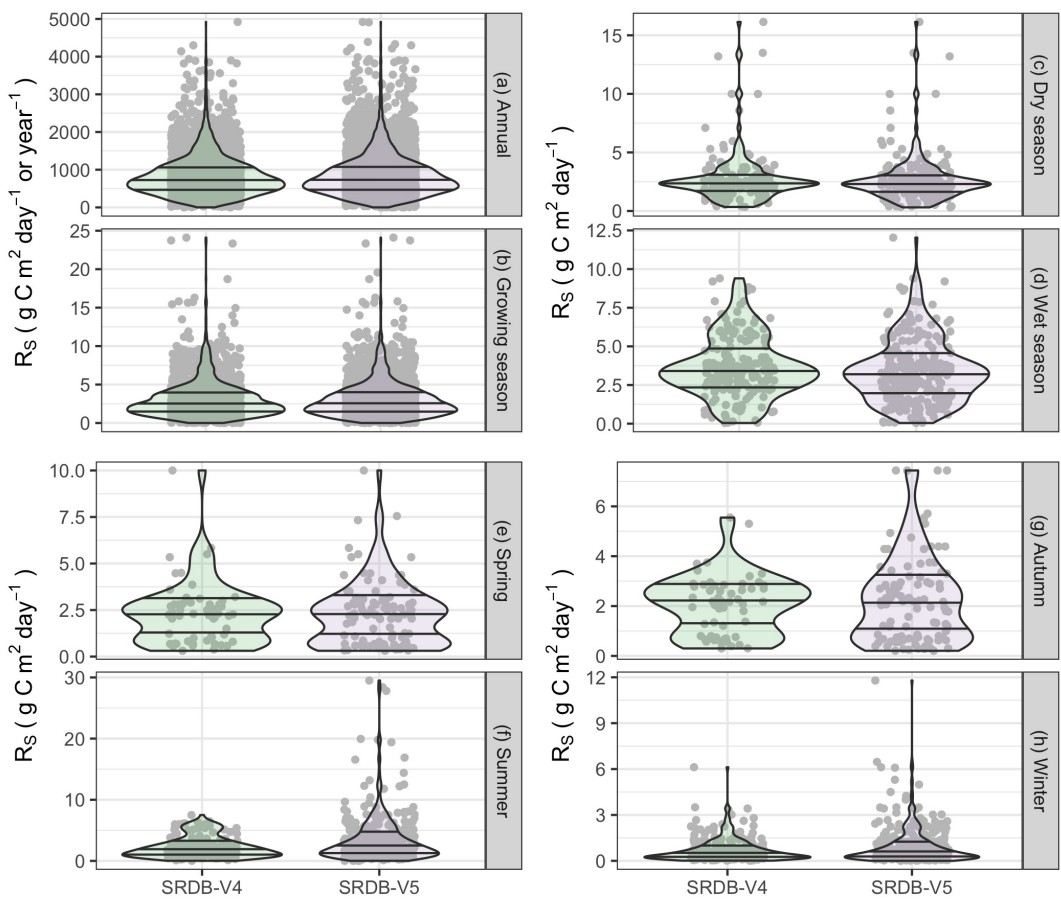


**Figure 5. Comparison of annual soil respiration ($R_S$) and seasonal $R_S$ (growing, dry, and wet seasons, spring, summer, autumn, and winter) observations from SRDB-V4 vs. that from SRDB-V5.** In summary, adding new measurements does not change the distribution of annual $R_S$ or seasonal $R_S$ in the databases.

## 4. Discussion

**4.1 Forecasting global $R_S$, $R_H$, and $R_A$**

**The updated SRDB-V5 provides opportunities for constraining global $R_S$ estimates in the future.** Currently, estimated global $R_S$ ranged from 68-101 Pg C yr$^{-1}$, with many uncertainties associated with measurements and propagation of errors evident when upscaling site-specific $R_S$ measurements to regional and global scales (Bond-Lamberty and Thomson, 2010b; Jian et al., 2018a, 2018b; Raich et al., 2002; Raich and Potter, 1995; Raich and
Schlesinger, 1992; Warner et al., 2019). For example, $R_S$ has been usually measured during daylight hours, implicitly assuming that measurements during this period represent the mean daily $R_S$. In a water-limited ecosystem, however, Cueva et al. (2017) estimated a time-of-day bias ranging from -29 to +40%. On the global scale, based on the HGRsD, Jian et al. (2018c) found that not measuring $R_S$ 24-hours continuously contributed less than 6% of bias when estimating diurnal $R_S$. Quantifying the amount of bias required detailed information about when $R_S$ was
measured and how long the measurement lasted (Jian et al., 2018c). In the SRDB-V5, we revised all the studies and



collected the "*Time_of_day*" information, which should enable future analyses of how $R_S$ measurement bias is related to when Rs measurements were collected.

**It is also widely accepted that chamber properties (e.g., volume, area)** (Davidson et al., 2002) **and collar insertion depth** (Heinemeyer et al., 2011) **affect the $R_S$ measurement accuracy**, but in global scale, this has not been quantitatively tested before to our knowledge. We added information in the SRDB-V5 to enable researchers to investigate whether chamber area (smaller chambers are more vulnerable to edge effects, while larger chambers may experience inadequate air mixing), collar height (which may affect air mixing in the chamber), and insertion depth (which may cut off roots) affect $R_S$ measurement accuracy and bias at seasonal to annual scales.


    **Comparing SRDB-V1 through SRDB-V5, we found that the uneven spatial distribution of $R_S$ sites has improved, but bias still remains, with measurements conducted unevenly around the world and in climate space (Figure 2-4).** The reason for the spatially-uneven coverage of $R_S$ sites is a combination of economy, national policy, environmental conditions, spatial heterogeneity, and many other issues. Most obviously, the northern
hemisphere has much more data than the southern hemisphere, as the most economically developed and wealthiest countries tend to be in the middle latitude of the Northern hemisphere, and thus more funds, infrastructure, and a broader and deeper pool of students and technical experts are all available to support on-site $R_S$ measurement in these regions.

**Improving modelling methods may help mitigate the uneven spatial distribution of $R_S$ sites.** For example, Jian et al. (2018b) found that how $R_S$ responds to temperature is significantly different among climate regions, and therefore climate-specific models may be more appropriate than a global single model to estimate global $R_S$. Alternatively, machine learning approaches that account for non-linearity and multiple potential combinations of environmental factors have been used to estimate global $R_S$ (Warner et al., 2019). SRDB-V5 also significantly
increased the $R_S$ sample size, and analyses could be conducted to test whether the increasing sample size of $R_S$ helps reduce uncertainty when upscaling from site to global scale $R_S$. We recognize that there are many other possible sources of bias, but it is nonetheless possible that the biogeochemistry community will be able to use SRDB-V5 to improve the confidence of global $R_S$ modeling and constrain global carbon cycle estimates.

**Linking SRDB-V5, MGRsD, HGRsD, and COSORE provides an opportunity for global $R_H$ and $R_A$ estimates.** Soil respiration mainly consists of two parts, $R_H$ and $R_A$, but it is difficult to separate these two components, and much less $R_H$ and $R_A$ data are available in the SRDB (Bond-Lamberty and Thomson, 2010a). Due to a lack of data, far fewer studies have analyzed $R_H$ and $R_A$ and estimated global $R_H$ and $R_A$ in the past decades. According to our knowledge, there are only five global $R_H$ (or $R_A$) estimates based on the very limited extant data (n < 500)
(Hashimoto et al., 2015; Konings et al., 2019; Tang et al., n.d.; Warner et al., 2019; Yao et al., n.d.). In the "srdb-equations" file, response of $R_H$ and $R_A$ to temperature and moisture information will be recorded, which will inspire the study of $R_H$ and $R_A$ and how they respond to temperature and soil moisture in the future. Further, we argue that a big advantage of global soil respiration databases with finer temporal resolution (i.e., MGRsD, HGRsD, and COSORE) is that the sample size of $R_H$ and $R_A$ could be greatly increased (e.g., sample size could be ten-fold
increased if using monthly time-scale). In addition, the spatial coverage of $R_H$ and $R_A$ data could also be improved, because sites not measured year-round $R_H$ and $R_A$ in SRDB (due to annual time-scale) could be compiled into MGRsD and HGRsD whenever $R_H$ or $R_A$ was measured. Based on the monthly $R_H$ and $R_A$ data and how they related to environmental conditions (such as temperature and precipitation), monthly global $R_H$ and $R_A$ products could be generated, which provide useful data products for the Earth System Models' (ESMs) benchmarking. The
disadvantages of the smaller timescale databases (MGRsD, HGRsD, and COSORE) is that those databases usually have much less spatial coverage; and much more data is available from the growing season than from the non-growing season. Therefore, spatial upscaling including time may result in additional bias and associated uncertainty that must be carefully investigated.

### 4.2 Perspective

**The updated SRBD-V5 will further support the analysis of how different manipulations affect $R_S$.** In the past decades, many field experiments have been conducted to study different questions, for example, how soil carbon responds to global climatic warming and changes in precipitation patterns (Vicca et al., 2014); or how human activities (forest management, agriculture cultivation, and pollution) affect terrestrial carbon cycling and soil carbon stock (Carrillo et al., 2014; Jasek et al., 2014). However, inconsistent results from different experiments have

generated debate regarding the effects of environmental factors and manipulations in $R_S$. Now SRDB-V5 includes $R_S$ measurements from both control and different kinds of treatments, providing opportunities for synthesis analysis of how manipulation affects $R_S$. However, these treatment data about $R_S$ measurements were rarely used in the past decade, as the manipulation information in older versions of SRDB was not standardized and thus could not easily be used. The updated and standardized SRDB-V5 manipulation codes have the potential to enable manipulation-

driven studies on the macro-to-global scale.

### 4.3 Future improvements

**We made an effort to resolve some issues in the old versions of SRDB (V1-V4), but the database needs to be continuously improved in the future.** There is much more potentially useful information that could be included in future SRDB updates, although it is important to remember that every additional piece of information comes with a

never-ending cost (in terms of data entry time, quality assurance/quality control, etc). 1) *Number_of_collar*: The number of collars within a certain study area is important information to evaluate the representability of the $R_S$ measurements; 2) Soil organic carbon (SOC) from regional or global estimates (Guevara et al., 2020; Hengl et al., 2017); 3) currently, *Site_ID* in SRDB-V5 are only comparable with *Site_ID* of MGRsD and HGRsD, further updates to *Site_ID* so it can connect with more external datasets [e.g., FLUXNET, COSORE, and AmeriFlux and a global

database of forest carbon stocks and fluxes (ForC) (Anderson-Teixeira et al., 2018b)]; *Annual_soil_moisture* to include a mean value of soil moisture or intra-annual soil variability derived from remote sensing (Guevara and Vargas, 2019) when this variable was not measured at the site . In addition, some meta information can be improved. For example, there are still 276 manipulation types in the SRDB-V5, and many manipulation types (n=96 out of 276) with only 1 row of records. Efforts could be made in the next version of database update to further simplify the

manipulation of SRDB. We recognize that with thousands of publications included in the SRDB, it is known that some entries are incorrect and some information may have been missed during litterature collection. In the past years, users have pointed out many data input errors and missing data issues on the SRDB, we made a great effort to check and many corrections have been made. However, it is inevitable that mistakes and missing information still exist, therefore, there is a pressing need to continue with the development of quality assurance and quality control

for each update.

### 4.4 Reducing interoperability barriers

**High interoperability is needed to maximize the benefits of SRDB-V5 to improve our understanding of the global carbon cycle.** Interoperability has been defined as an organized collective effort with the ultimate goal to maximize sharing and using information to produce knowledge; and high interoperability is achieved by reducing

conceptual, technological, organizational and cultural barriers (Vargas et al., 2017). The improved SRDB-V5 has reduced conceptual barriers as it provides a standardized and replicable framework to organize global $R_S$ information that has been used for over a decade (Bond-Lamberty and Thomson, 2010a). It has reduced technological barriers by improving standardization of data fields (see Tables 1-3), data formats compatible with other databases, and providing flexible R scripts in a Github repository for end users and potential data contributors.

We recognize that measuring $R_S$ has other technological barriers (e.g., standardization of instrumentation, electrical power supply) that limit the collection of new measurements in harsh environments or wide implementation in developing countries. Organizational barriers remain a challenge as this is a bottom up effort in need of long-term support to continue improving the quality and developing the new versions of the SRDB. Finally, we believe that





cultural barriers have been reduced as the global scientific community has improved in recognizing the importance
      of standardized databases and data sharing following FAIR principles.

**5. Data availability**

Findability and Accessibility were well considered and described when SRDB-V1 was published (Bond-Lamberty
and Thomson, 2010a). To summarize the updating progress, SRDB-V1 was the first availability of the full data set,
released on 2010/05/28; SRDB-V2 was released on 2012/03/13, $R_S$ date of publications from 2011 was integrated
into the database; SRDB-V3 was released on 2014/08/04, $R_S$ data of literature from 2012 was collected and added;
      SRDB-V4 was released on 2018/11/21, $R_S$ data of literature through 2015 were collected and compiled into the
      database; and SRDB-V5 was released on 2020/04/24, $R_S$ data of literature from 2017 was collected and added (Jian
      and Bond-Lamberty, 2020). The version release information was recorded at the Oak Ridge National Laboratory's
      Distributed Active Archive Center ORNL-DAAC (Submitted). All data and code to reproduce the results in this
study can be found at: Jian, Jinshi, Bond-Lamberty, Ben. (2020). jinshijian/ESSD: SRDB-V5 first release (Version
      v1.0.0) [Data set]. Zenodo. http://doi.org/10.5281/zenodo.3876443.

**Conclusion**

A global soil respiration database (SRDB) was developed to integrate soil respiration measurements from the globe
a decade ago. Since the first release in 2010 (SRDB-V1), it has been widely used to advance our understanding of
carbon decomposition related questions. Here, we restructured SRDB to a new version (SRDB-V5) following FAIR
      principles. We show that the SRDB substantially improved its representativeness compared with the older versions
      (SRDB-V1 to SRDB-V4, Figure S1 and S2) and improved its spatial coverage. A primary goal of SRDB-V5 is to
      improve the interoperability and reusability, and make it possible for scientists to contribute in the future with the
      ultimate goal to improve our understanding of the global carbon cycle. With those goals in mind, the revised SRDB-
V5 is now more user-friendly for the ecology, biogeochemistry, and modeling communities.

**Acknowledgments**
This research was supported by the US Department of Energy, Office of Science, Biological and Environmental
Research as part of the Terrestrial Ecosystem Sciences Program. The Pacific Northwest National Laboratory is
operated for DOE by Battelle Memorial Institute under contract DE-AC05-76RL01830. We would like to thank
      Dalei Hao for his help with the MODIS landuse data processing. R.V. acknowledges support from NASA CMS
      grant number 80NSSC18K0173.

**Author contributions**
B.B.-L and J.J. designed the new version global soil respiration database (SRDB-V5). B.B.-L searched and
      downloaded the new papers until 2017 and compiled the meta-information. B.B.-L, M.H., R.M., J.M., D.P. and J.J.
      contributed to data collection, N.K. collected data in Russian, K.A.T. and V.H. raised many useful suggestions while
      working to integrate with ForC, R.V. and E.S. provided feedback and insights in all phases. J.J. wrote the
      manuscript in close collaboration with all authors.

**Competing interests**
The authors declare no competing interests.

**Using and citing SRDB-V5**
SRDB-V5 can be used for individual, academic, research, commercial, and other usage, and can be repackaged
      without written permission. Research and non-research products using SRDB-V5 should cite this publication.



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
