# Peer review of "A restructured and updated global soil respiration database (SRDB-V5)"

_Earth System Science Data, 2020_

## Referee Comment (RC1) · Anonymous Referee #1 · 9 Sep 2020

General comments: The authors report the update of global soil respiration database (SRDB), which has been used in soil carbon studies and has played important roles in the research field. The update is not only increased records but also some restructuring of the database. The paper clearly describes the update of the database and changes in data base structures. I recommend the publication of this paper.

Below are some minor comments.

Line 196: Table 2, 3, "Number of rows" "Number of rows" sounds very operational term. Should be replaced with "records" or something.

Line 231: "did not change the distribution of annual Rs" I agree, but how about regional averages (e.g. south America, Africa etc.)? I guess in some areas, the new records in

[Figure]

SRDB-V5 substantially increased the number of observations and probably changed the annual Rs average and distribution.

Line 325: "Tang et al. n.d., Yao et al. n.d." Please include the year for these two papers.

Line 371: "interoperability" Does your team have any plan to start "data standardization, submission/acceptance, curation, data base construction, and distribution of raw data" platform for soil respiration, like Fluxnet?

Line 379: "providing flexible R scripts in a Github repository for end users". It is not clear that the R scripts are for what?

Fig. 1 Can you improve the appearance of this figure?

---

## Author Comment (AC1) · 21 Oct 2020

Hello dear reviewer,

Thank you for your overall positive assessment and very useful comments on our manuscript. Below please see our response to your comments.

Thanks

Line 196: Table 2, 3, "Number of rows" "Number of rows" sounds very operational term. Should be replaced with "records" or something. Response: We agree, will change it to "number of records".

Line 231: "did not change the distribution of annual Rs" I agree, but how about regional

averages (e.g. south America, Africa etc.)? I guess in some areas, the new records in SRDB-V5 substantially increased the number of observations and probably changed the annual Rs average and distribution. Response: We will change the topic sentence accordingly.

Line 325: "Tang et al. n.d., Yao et al. n.d." Please include the year for these two papers. Response: Thank you for pointing out this, we will update the citation and add the year information.

Line 371: "interoperability" Does your team have any plan to start "data standardization, submission/acceptance, curation, data base construction, and distribution of raw data" platform for soil respiration, like Fluxnet? Response: This is a great idea, unfortunately, this have not been done in this version of SRDB, but maybe considered in the next version of updating.

Line 379: "providing flexible R scripts in a Github repository for end users". It is not clear that the R scripts are for what? Response: We will describe the R script in more detail in the revision.

Fig. 1 Can you improve the appearance of this figure? Response: I am not sure whether you are talking about Figure 2, as we think Figure 1 is pretty straightforward and clear. If you are talking about Figure 2, we actually tried multiple ways and found the current version of Figure 1 is the best, but we can try to make the background color lighter and see whether it is better.
* * *

---

## Referee Comment (RC2) · Anonymous Referee #1 · 23 Oct 2020

Thank you for your reply to the comments.

What I didn't like about Fig. 1 is 1) the fonts are too small. 2) the colors are too strong.

I believe point 1 must be solved, but point 2 is up to you.

Thank you.

—————————————————————

---

## Referee Comment (RC3) · Anonymous Referee #2 · 30 Nov 2020

This is a data release of an update to a database of global soil respiration measurements, the first version of which was released 10 years ago. Since then, 4 updates have been made to the database. This most recent update was significant, changing the structure of the database and adding or changing many fields, thus warranting this data release paper. This paper very clearly documents the background of the original database, past utility and importance of the SRDB, the justification for a significant update, and the future potential usage of the newest update to the SRDB. I offer comments below in hope this can be used to further improve the paper.

Line 76: I would add "each year" to the end of the sentence ending in "its use continues to increase" 115: One detail that is not addressed is the file format. Has the file format of the SRDB changed (it appears not)? A reader might wonder if any changes have

occurred in terms of the file format. Please add some text that indicates what the original file format is and why it was kept this way or changed in the new version.

130: "had not been used" - it is unclear exactly what this means. If zero papers have reported these metrics in the past 10 years, please state that.

160 - Methods in general: Metadata on latitude and longitude of locations would be extremely helpful for those hoping to link soil respiration measurements to spatial data. Please consider adding this, or if not, please provide justification why this was not considered. My main concerns are: Studies reporting lat/lon at different levels of precision (i.e., decimal points), but the implied precision in the database might not actually reflect what was recorded. Studies using different methods for recording lat/lon - GPS units may have wide variation in spatial accuracy. Most studies may not report spatial accuracy/precision. Reporting one general lat/lon for the study site versus lat/lon for the individual study sites. How is this handled? This is a significant issue for linking up to spatial data.

Figured 3 and 4: I understand the utility of density distribution plots and use them often myself, but they do not convey any information about the number of observations. I think there are 2 ways the paper can improve in this respect: Offer explanation in the Figure captions about what the density plots represent, and provide the number of observations for each category presented.

---

## Author Response (AR1)

**Reviewer 1**

General comments: The authors report the update of global soil respiration database (SRDB), which has been used in soil carbon studies and has played important roles in the research field. The update is not only increased records but also some restructuring of the database. The paper clearly describes the update of the database and changes in data base structures. I recommend the publication of this paper.
Response: Thank you for your overall positive assessment and very useful comments on our manuscript. Below please see our response to your comments. Thanks

Below are some minor comments.

Line 196: Table 2, 3, "Number of rows" "Number of rows" sounds very operational term. Should be replaced with "records" or something.
Response: We agree, changed to "Number of records".

Line 231: "did not change the distribution of annual Rs" I agree, but how about regional averages (e.g. south America, Africa etc.)? I guess in some areas, the new records in SRDB-V5 substantially increased the number of observations and probably changed the annual Rs average and distribution.
Response: Revised as: "Adding new measurements in SRDB-V5 has substantially increased total observations and the spatial coverage of sites was improved compared with SRDB-V4 (Figure 2). However, the distribution of annual RS seasonal RS (growing, dry, wet, spring, summer, autumn, and winter season RS) were similar in the SRDB-V5 compared to SRDB-V4 (Figure 5)."

Line 325: "Tang et al. n.d., Yao et al. n.d." Please include the year for these two papers.
Response: Thank you for pointing out this, we added year information for Tang et al (2020); we deleted Yao et al. because this paper is not publicly available yet.

Line 371: "interoperability" Does your team have any plan to start "data standardization, submission/acceptance, curation, data base construction, and distribution of raw data" platform for soil respiration, like Fluxnet?
Response: This is a great idea, unfortunately, this have not been done in this version of SRDB, but maybe considered in the next version of updating.

Line 379: "providing flexible R scripts in a Github repository for end users". It is not clear that the R scripts are for what?
Response: This was described in section 2.4, we mentioned it in the revision now.

Fig. 1 Can you improve the appearance of this figure?
Response: Figure 1 re-plotted.

**Reviewer 2**
This is a data release of an update to a database of global soil respiration measure- ments, the first version of which was released 10 years ago. Since then, 4 updates have been made to the database. This most recent update was significant, changing the structure of the database and adding or changing many fields, thus warranting this data release paper. This paper very clearly documents the background of the original database, past utility and importance of the SRDB, the justification for a significant update, and the future potential usage of the newest update to the SRDB. I offer com- ments below in hope this can be used to further improve the paper.
Response: Thank you very much for your overall positive assessment about our manuscript.

Line 76: I would add "each year" to the end of the sentence ending in "its use continues to increase"
Response: We added "each year" to the end of this sentence.

115: One detail that is not addressed is the file format. Has the file format of the SRDB changed (it appears not)? A reader might wonder if any changes have occurred in terms of the file format. Please add some text that indicates what the original file format is and why it was kept this way or changed in the new version.
Response: Good point. We now specify that the file format was not changed.

130: "had not been used" - it is unclear exactly what this means. If zero papers have reported these metrics in the past 10 years, please state that.
Response: That is correct, and we added "(according to our search, data from these four columns have never been used)" to explain this in the revision.

160 - Methods in general: Metadata on latitude and longitude of locations would be extremely helpful for those hoping to link soil respiration measurements to spatial data. Please consider adding this, or if not, please provide justification why this was not con- sidered. My main concerns are: Studies reporting lat/lon at different levels of precision (i.e., decimal points), but the implied precision in the database might not actually re- flect what was recorded. Studies using different methods for recording lat/lon - GPS units may have wide variation in spatial accuracy. Most studies may not report spatial accuracy/precision. Reporting one general lat/lon for the study site versus lat/lon for the individual study sites. How is this handled? This is a significant issue for linking up to spatial data.
Response: We added a paragraph (lines 215-226) to talk about this in the revised manuscript. In generally, the SRDB is limited by the precision and accuracy of its study inputs, and when users are working with very high-resolution data, we agree that this issue needs to be considered carefully.

Figured 3 and 4: I understand the utility of density distribution plots and use them often myself, but they do not convey any information about the number of observations. I think there are 2 ways the paper can improve in this respect: Offer explanation in the Figure captions about what the density plots represent, and provide the number of observations for each category presented.
Response: We added number of observations in each panel of Figure 3 now. We did not add these numbers in Figure 4 because the sample size is the same as Figure 3, we clarified this in the figure caption of Figure 4.

[revised manuscript text omitted]